# 'They eat it like sweets': A mixed methods study of antibiotic perceptions and their use among patients, prescribers and pharmacists in a district hospital in Kabul, Afghanistan

**Doris Burtscher**[1]*, **Rafael Van den Bergh**[2], **Masood Nasim**[3], **Gbane Mahama**[2], **Sokhieng Au**[2], **Anita Williams**[4,5], **Abdul Sattar**[6], **Suzanne Penfold**[7], **Catherine Van Overloop**[2], **Sahar Bajis**[8]

**1** Vienna Evaluation Unit, Médecins Sans Frontières, Vienna, Austria, **2** Operational Centre Brussels, Médecins Sans Frontières, Brussels, Belgium, **3** Médecins Sans Frontières Afghanistan, Kabul, Afghanistan, **4** Operational Research (LuxOR) Unit, Médecins Sans Frontières, Luxembourg, Luxembourg, **5** Middle East Medical Unit (MEMU), Médecins Sans Frontières, Beirut, Lebanon, **6** Ahmad Shah Baba Hospital, Ministry of Public Health, Kabul, Afghanistan, **7** Independent Public Health Research Consultant, Trnava, Slovakia, **8** The Kirby Institute, UNSW Sydney, Sydney, Australia

* doris.burtscher@vienna.msf.org, doris.burtscher@gmail.com

**Data Availability Statement:** Data cannot be shared publicly because to protect any individual

## Abstract

### Background

Antibiotic resistance is a growing public health threat. In Afghanistan, high levels of indiscriminate antibiotic use exist, and healthcare programmes are not informed by understanding of local attitudes towards rational antibiotic use. Médecins Sans Frontières is an international non-governmental organization providing healthcare services to the Ahmad Shah Baba (ASB) District Hospital in Kabul, Afghanistan, since 2009. This mixed-methods study aimed to explore the perceptions and attitudes toward antibiotics among patients, prescribers, and pharmacists in the ASB District hospital outpatient department.

### Methods and findings

Knowledge of antibiotics including their purpose and function, how and why they are used, and drivers for choice of antibiotic was examined at patient, prescriber, and provider-level. The first phase of the study, an exploratory qualitative component using an interpretative approach, was used to inform the second phase, a structured survey. Thirty-six interviews were conducted with 39 participants (21 patients or caretakers and 18 hospital health workers). Three hundred and fifty-one (351) patients and caretakers completed the second phase, the structured survey. This study found that poor knowledge of antibiotics and antibiotic resistance is a driving factor for inappropriate use of antibiotics. Participant perceptions of living in a polluted environment drove the high demand and perceived 'need' for antibiotics: patients, doctors and pharmacists alike consider dirty and dusty living conditions as causes of 'disease' in the body, requiring antibiotics to 'clean' and 'strengthen' it.

that has particpated in the interviews. Data contain very personal information. Data are availabel from the MSF Institutional Data Access Committee (contact via annick.antierens@brussels.msf.org) for researchers who meet the criteria for access to confidential data.

**Funding:** The author(s) received no specific funding for this work.

**Competing interests:** The authors have declared that no competing interests exist.

## Conclusions

Findings highlight the need for strategies to improve awareness and knowledge of the general public, improve practice of doctors and pharmacists, regulate antibiotic dispensing in private pharmacies, and implement antibiotic stewardship in hospitals.

## Introduction

Globally, an estimated 700,000 people die each year due to drug-resistant strains of common bacterial infections, human immunodeficiency virus, tuberculosis and malaria [1]. Without effective action, the number of deaths due to drug-resistant infections is projected to rise to 10 million per year by 2050 [2]. Irrational use of antibiotics in healthcare is a key driver of antibiotic resistance and has been associated with poor patient outcomes and wasted resources [3–8]. Barriers to rational use of antibiotics include lack of awareness by healthcare providers and patients, illegal or unregulated antibiotic supply, circulation of substandard or expired antibiotics, and lack of access to healthcare [9]. While prescribing practices of healthcare providers have been shown to account for a substantial portion of irrational antibiotic use globally, patient perception, health-seeking behaviour and choice of medicines represent strong determinants of antibiotic use [5, 10–12]. Efforts in curbing irrational use of antibiotics must consider public attitudes and expectations.

Self-medication studies in diverse settings have demonstrated that the use of antibiotics was determined mainly by previous experience with efficacious treatment, perception of antibiotics as strong and effective drugs, availability of over-the-counter antibiotics, perceived severity of sickness, and lack of information regarding antibiotics' characteristics and the issue of resistance [13–20]. Patients' perceptions and experience have been shown to influence patient-provider dynamics, often determining whether antibiotics are ultimately prescribed and how they are administered, with patients sometimes exaggerating their symptoms in order to get antibiotics [21–23].

Understanding local attitudes to antibiotic use is needed to develop appropriate and adapted antibiotic stewardship programs (i.e. strategies to reduce irrational antibiotic use) [24]. A previous study conducted at the Ahmad Shah Baba Hospital (ASB), Kabul, Afghanistan, reported regular over-prescription of antibiotics, despite programmatic measures implemented to improve prescribing practices [25]. Very few studies have examined antibiotic use among local communities in Afghanistan [25–28]; most studies in the country have focused on antibiotic use and resistance among international armed forces [29–31], and little or no attention has been devoted to the Afghan population. Furthermore, major gaps exist in knowledge of the experiences and attitudes towards antibiotic use among patients, prescribers, and pharmacists in Afghanistan.

The aim of this mixed-methods study was to explore the knowledge, perceptions and attitudes toward antibiotics among patients, prescribers, and pharmacists in a district hospital in Kabul, Afghanistan. Specifically, the study aimed to examine: 1) knowledge of the purpose and action of antibiotics, 2) drivers of choice, 3) how antibiotics were used and 4) rationales for use among patients, prescribers and pharmacists.

## Methods

### Study design

This was a mixed-methods study conducted in two phases: first an exploratory qualitative component using an interpretative approach, followed by a survey developed from the

preliminary findings of the qualitative component of the study. Qualitative data was collected between September 1st to October 1st, 2014, and the survey was conducted between 19th October and 10th November 2014.

## Setting

Médecins Sans Frontières (MSF), an international non-governmental organization, provided healthcare services support to the ASB District Hospital in Kabul, Afghanistan, from 2009 to 2019. The ASB Hospital had a capacity of 69 beds with surgical, maternity, paediatric, and internal medicine inpatient wards. The study was conducted in the male and female outpatient departments (OPD) of the ASB hospital in 2014. In that year, the number of public health facilities in the district were not enough to cover the total population; gaps in public healthcare were being filled by private clinics–there were eight private hospitals, 103 private clinics, and 240 drug dispensaries.

The OPD of ASB was one of the busiest departments in the hospital, averaging 7,500 consultations a month. Services were provided by eight doctors and one OPD medical supervisor. Medical staff at ASB underwent education and training on MSF clinical guidelines and good prescribing practices, including rationale use of antibiotics, however, uniform education intervention focused on rational use of antibiotics was not implemented during the study period.

Patients attending the OPD received their prescribed medications from the dispensary located inside the hospital premises, which was supplied by the MSF ASB central pharmacy. Health promoters provided health education to patients and caretakers in the female and male waiting areas on a variety of topics, including appropriate antibiotic use. All services in the hospital were provided free of charge.

## Qualitative component

**Study participants and sampling.** The exploratory qualitative component used an interpretative approach [32]. Individual in-depth interviews and group discussions were conducted, with participants choosing if they would like to be interviewed alone or with their colleagues or acquaintances. Eligible participants were adult patients (≥18 years), or parents or caretakers of children attending the OPD. Medical and allied health staff in the OPD involved in the prescription pathway including doctors, pharmacists and dispensers, and health promoters were also eligible. Within the staff and patient/caretaker groups a purposive sampling technique was used to ensure representation of the following categories: age, sex, education level, setting, and ethnicity. Data were collected until no additional insights related to the research questions were gained (thematic saturation) [33].

**Participant recruitment.** In September 2014, participants were recruited from the female triage, female consultation and female caretaker area, male consultation and male caretaker area, vaccination, the OPD dispensary, and two private pharmacies located in the building next to the hospital. No recruitment was conducted in the male triage due to a low number of male patients at the time of recruitment. Patient and caretaker participants were approached before their consultation while they were waiting for their appointments. Health staff, including medical staff and health promoters were informed about the study and its purpose; staff participants were interviewed during breaks from their duties. There were no refusals to participate.

**Data collection.** The interviews and discussions were conducted by author DB, who is a female medical anthropologist. Interviews and discussions were conducted in Pashto or Dari by two translators from September 7–29, 2014. Careful explanations of the role of the PI and

her neutrality and strict assurance of anonymity and confidentiality were given for each interviewee. All interviews were audio-recorded.

The in-depth interviews were guided by topic-led questions. Interviews with patients and caretakers were organized through the translators and with the help of the health promoters (HP) in the waiting area of the outpatient department (OPD) and triage. To maintain participant confidentiality and privacy, interviews were conducted in a separate room. Interviews with the health staff also occurred in this separate room and were performed during their breaks.

The questions asked of patients were theme-based related to health-seeking behaviour, knowledge and perceptions of medications in general and of antibiotics, experiences with medication use and antibiotics, and prescriber/provider-patient/caretaker relationships and experiences. The themes of interviews with prescribers and pharmacists explored health seeking behaviour, knowledge and experience with antibiotics, prescribing practices, experiences with patient's expectations, and relationships with patients. The question guide is available as S1 Appendix.

Following qualitative interview procedures, question guides were kept flexible; the order of questions was driven by the nature of each participant's answers, which means that both the wording of questions and the order they were asked during interviews were frequently modified.

**Data analysis.** All interviews were translated and transcribed into English. A memo was written for every interview to summarise what was said for each theme. The memos formed the basis of the qualitative data analysis using NVivo v10 (QSR International, Australia). After the initial coding of the memos, results were labelled with age, gender, education, and participant type (patient, caretaker or staff with position specified if staff) and entered into a code report. The manual analysis of these memos was based on a qualitative content analysis by Mayring [34]: transcribed content was coded according to themes arising and analysed by simply describing the characteristics of the data, and interpretively by focusing on what might be meant by the responses [34, 35]. Empirical data were analysed in an inductive way [32]. Continuous reflection on the data was part of the creative process of the analysis, and necessary for contextualizing and linking findings to anthropological theory. Validity of data was ensured by a 'thick' description of the research context [36]. Each step of the above-mentioned coding and analysis was performed by author DB and a volunteer. All data were stored without identifying information.

## Quantitative component

**Study participants and recruitment.** Adult patients or caretakers ≥18 years present in the waiting rooms of the OPDs, vaccination and dispensary areas were invited to complete the survey four days a week between October 10 and November 11, 2014. Potential participants were selected by quota sampling to aim for approximately equal numbers of male patients, female patients, and caretakers (regardless of gender), with a minimum of 12 participants per translator per day approached.

**Data collection.** The survey consisted of 19 questions developed from the findings of the qualitative interviews relating to participant knowledge and views of antibiotics, expectations of drug receipt at the clinic, previous sources and modes of antibiotic use. Surveys were translated into Pashto and pre-piloted in the population. Study translators administered the survey by reading it aloud to the study participants; participants then either selected the option for which the translator ticked the relevant box, or, if the participant gave an answer outside these options, the translator wrote the exact response in a free-text box. The questionnaire is available as S2 Appendix.

**Sample size.** The sample size calculation assumed 30% prevalence of the conditions of interest (e.g., specific knowledge of antibiotics, attitudes to antibiotic use etc.) With a 5% confidence interval and 95% confidence level this meant a minimum of 315 participants needed to be interviewed.

**Data analysis.** Answers from the survey were double entered into an EpiData v2.0 (EpiData Association, Denmark) database and cross-checked. Data were analysed descriptively using Microsoft Excel calculating the number and proportions for responses to categorical answers or median with IQR for numerical answers.

## Ethics

The study was approved by the Institutional Review Board of the Afghanistan National Public Health Institute, the Ministry of Public Health Afghanistan in Kabul, Afghanistan, and by the Ethics Review Board of MSF (ID 1406). Oral and written informed consent was obtained from all participants of the study by the translators, who had been trained in informing study participants and consent seeking. The patient information and consent form were translated into Dari and Pashto. All interviewed individuals were asked beforehand where they preferred to be interviewed. A separate room, where confidentiality could be maintained, was available.

## Results

### Study participants

A total of 36 interviews with 39 participants (21 patients/caretakers and 18 healthcare workers) were conducted. The healthcare workers interviewed were seven doctors, six health promoters and five pharmacists. Nearly all participants had individual interviews; two caretakers and four health promoters were interviewed in pairs. Patients and caretakers were most commonly aged less than 30 years, of Pashto ethnicity and illiterate (Table 1). An approximately equal number of men and women were interviewed.

A total of 351 participants completed the survey, with all patients and caretakers approached consenting to participate. The median age of participants was 28 years (IQR 21–36), and 60% were female (Table 2). The majority of participants resided in an urban setting (98%). Participants had a median of three children (IQR 0–6) and lived in a household of median nine people (IQR 7–12).

Responses to all survey questions are available in the S1 Table.

**Table 1. Characteristics of patient and caretaker interview participants (n = 21).**

| Participant characteristics | Patients | Caretakers | Total |
|---|---:|---:|---:|
| Age | | | |
| <20 years | 3 | 3 | 6 |
| 20–29 | 3 | 6 | 9 |
| 30–39 | 1 | 2 | 3 |
| 40–49 | 1 | 1 | 2 |
| ≥50 years | 1 | 0 | 1 |
| Sex | | | |
| Males | 4 | 6 | 10 |
| Females | 5 | 6 | 11 |
| Total | 9 | 12 | 21 |

**Table 2. Socio-demographic characteristics of survey participants (n = 351).**

| Participant characteristics | n | % |
|---|---|---|
| Participant type* | | |
| Adult patient | 239 | 68 |
| Caretaker | 112 | 32 |
| Sex | | |
| Male | 142 | 40 |
| Female | 209 | 60 |
| Age (years) | | |
| <20 | 53 | 15 |
| 20–29 | 130 | 37 |
| 30–39 | 87 | 25 |
| 40–49 | 43 | 12 |
| > = 50 | 34 | 10 |
| Not recorded | 4 | 1 |
| Age of children of caretakers† | | |
| <5 | 43 | 38 |
| > = 5 | 28 | 25 |
| Not recorded | 41 | 37 |
| Area of residence | | |
| Urban | 343 | 98 |
| Rural | 8 | 2 |
| | **Median** | **IQR** |
| Number of children | 3 | (0–6) |
| Household size | 9 | (7–12) |

*Primary participant type. It should be noted that some participants came to the clinic as both patients and caretakers.

†n = 112 for the age of children of the caretakers who answered this survey.

## Perceptions and knowledge of antibiotics

**Cleaning the body and the blood from dirt.** Many patients and caretakers mentioned antibiotics by name or used words which evoked an understanding of medicines acting against dirt and microbes e.g., "zedi cherk" (anti-dirt, anti-microbe). Sixty-five percent of survey participants responded that they had heard of antibiotics. The majority of participants agreed with statements about antibiotics being used to kill microbes (64%), are the best to treat infections (59%), and thought antibiotics can cure illness quickly (56%). Doctors reinforced this finding in the interviews by explaining how patients asked for "powder syrup", "orange pills" or simply "SIX capsules" (Male Doctor, 40 years).

The words "anti-dirt" and "anti-microbe" in local terms imply that the medication was supposed to clean the body from dirt, i.e., microbes. The dirty and dusty environment was believed by patients and caretakers to cause 'disease' in the body, including fertility issues. Doctors and pharmacists both referred to the lack of hygiene, pollution, dirt, and perceived impurity of the society they live in as reasons why people used antibiotics.

"We use 'zedi cherk' [anti-microbe] to kill the blood microbes so that she can get pregnant. Sometimes it is used to heal that disease which we get from cows and animals and we use

'zedi-cherk' medicine against blood microbes. When you do not have clean blood, you cannot get pregnant." (Male caretaker, 40 years)

**Powerful, famous, and rapid-acting medication.** Amoxicillin for adults or powder syrup (Amoxicillin) for children was the antibiotic most commonly mentioned in the interviews. Patients and caretakers favoured this antibiotic because they had used it themselves, had heard that it is a strong and effective medication, had received a recommendation from others, or had had it prescribed by a pharmacist or doctor. Most importantly, antibiotics were believed to cure illnesses 'quickly'. Some patients and caretakers could only name two medicines, and these were paracetamol and amoxicillin.

"They take most often Amoxicillin and Amoxil or Ampiclox, they ask a lot for, and they are famous." (Male HP, 40 years)

"Whenever we feel pain in our body, we take an antibiotic and it affects very quickly and is effective to our body. [. . .] Because the people know that it cures quickly; even those people from the village when they get antibiotics it cures them; they frequently ask for the same medications." (Female patient, 27 years)

In the interviews, patients and caretakers believed that a higher number of different drugs was better, but, above all, an injection was stronger than pills. Half of the respondents to the survey agreed that "more pills cure faster" (50%, n = 174) or that "different pills together cure faster" (52%, n = 182), whilst 84% of participants agreed that injections/serums (intravenous line) cure faster (n = 296). The vast majority of participants agreed that "quality is more important with quantity" for medications (97%, n = 342).

"When a patient has bad health condition and when you give him/her an injection it means you give him/her the entire world as a gift." (Male doctor, 39 years)

The idea of using antibiotics to protect and strengthen the body was common among patients and caretakers, with doctors confirming that patients wanted to take antibiotics because they thought it was the only way to keep them healthy. Antibiotics were believed to cure internally-caused malaises: a weak immune system and general body weakness were mentioned as reasons to use antibiotics.

"The antibiotic is making the body powerful against disease. (. . .) When bacteria enter to our body it will affect different parts of the body but when we use antibiotics it will easily remove it and will increase the level of efficiency in our body." (Male caretaker, 25 years)

**Sources of information and medication.** Participants explained that they knew where to get antibiotics and that it was easy to obtain them; when they did not have antibiotics at home or could not obtain antibiotics through medical consultation, participants went directly to the pharmacy for medication and relied on the vendor's recommendation. The questionnaire data corroborate this; 87% of participants indicated that they sourced medications from private pharmacies (n = 305). Participants explained that the pharmacist gave them antibiotics with or without a doctor's prescription.

"I ask the pharmacist and he gives me that [antibiotic], when I take that my disease gets cured, [. . .] the pharmacist knows what I need; he knows." (Male patient, 24 years)

A subset of patients and caretakers agreed with the statement that antibiotics must be taken according to the doctor's prescription, and incorrect use may cause adverse health outcomes. Participants in the questionnaire responded that either the doctor (86%, n = 301), or the dispenser (72%, n = 253) explains how to use the drugs. However, numerous participants in the interviews did not fully understand the concept of antibiotic resistance but acknowledged that antibiotics may have risks or a negative impact on the body.

## Patients' expectations of receiving antibiotics

Patients saw antibiotics as the 'desired medication' and expected to receive them from doctors. However, this expectation was discussed more by doctors than by the patients and caretakers in the interviews. Health promoters confirmed these demands for antibiotics as they often heard patients "insist" on antibiotics from the doctors. The survey found almost all respondents expected at least one medication to be dispensed as a result of their clinic visit (97%, n = 342), though, the majority of participants responded that "the doctor decides" which medication they would receive (76%, n = 266).

> "Two years ago, I have been eye witness of some case that when they [patients] don't get their desired medication [antibiotic] from male doctors, they were coming to the female doctors in order to get their desired medication (Male HP, 48 years)

Pharmacists who dispensed medication spoke about patients' and caretakers' anger when they realized that the doctor had prescribed other medications rather than antibiotics. Half of the participants indicated they would accept the decision of the doctor (51%, n = 141), with only 4% of participants saying they would complain (n = 15).

> "They [patients] think if we give them antibiotics, their pain would perfectly get cured, we have faced even those people who threw the medicine [not an antibiotic] on our face and they say: 'Take this medicine [back] we will not be cured with this.' Some people even threw stones on us." (Female Pharmacist, 50 years)

Doctors working in ASB hospital are obliged to comply with MSF prescribing guidelines, however, they also work in their private clinics, and reported feeling pressured to meet patients' expectations and to deal with patient frustrations at not receiving antibiotics. The fear of losing patients by refusing to prescribe antibiotics was considerable. Forty percent of survey participants indicated they would seek medications elsewhere if they do not receive the drugs they expect (n = 141), with 74% saying they would attend a private clinic (n = 105). When doctors followed MSF guidelines and significantly reduced the prescription of antibiotics, patients and caretakers doubted the motives and practice of MSF doctors and pharmacists (dispensers) concerning the drugs:

> "They [patients] think the doctors are working for their own benefit rather than paying attention to their health condition. They say the doctors want to have a stock of antibiotics in the hospital and for this the patients get a bad image of the doctors. The patients say the drugs are coming from foreign people [MSF] and you are jealous because you don't give them to us." (Male doctor, 39 years)

Patients also complained that doctors did not take enough time to talk to them and did not show enough empathy; patients expected doctors to pay attention to them as a people and not as "cases of illness". Patients and caretakers said that receiving good words and good drugs

from medical staff were equally important for them, but, when asked further about the subject, they clearly indicated that they placed more value on receiving medication. In an informal discussion with an MSF employee, he recounted that people said: ". . . a good doctor gives more time and more drugs" and "he [the doctor] has checked me with many machines". Both staff and patients/caretakers indicated that when patients had come from remote or deprived areas, antibiotics were prescribed more readily:

> "First, I try to convince them [that an antibiotic is not needed] and when the patient asks me again and when they have an infection in the throat or they have bloody diarrhoea or when they are from far, then we prescribe them antibiotics." (Male Doctor, 48 years)

Of the survey participants, only 2% of respondents were from rural areas.

From the doctors' perspective, in general, patients expected to receive antibiotics whenever they saw a doctor, one doctor explaining the "universal" use of antibiotics in Afghanistan as "they eat it like a sweet."

**Perceived risks and side effects of antibiotics.** The idea that medication in general, and antibiotics in particular, have side effects or may cause some risks did not affect the positive perception of antibiotics. Other participants talked about the side effects when taking antibiotics as in their opinion "every medication has certain side effects" and mentioned that using too many antibiotics over the course of several years can present negative effects. Twenty-three percent of survey respondents said that they sometimes stopped taking the antibiotics before the course was finished (n = 82). Of those 23%, 13% stopped early due to self-reported side effects (n = 11), and six percent due to a bad taste or smell (n = 5).

Some participants did not see any risks in taking antibiotics; in contrast, they believed the medication helped strengthen the body:

> "Antibiotic in general is a good medication; there is no risk while taking antibiotics. My classmate was a body builder and he used antibiotics, now his body gets enlarged day by day." (Male caretaker, 25 years)

## Antibiotic (mis)use

**Antibiotic use for inappropriate conditions.** Patients and caretakers alike mentioned consistently using antibiotics for various conditions, including common colds, infertility, general body pain and after a delivery, among others. The idea that antibiotics could be used to treat different conditions was also observed in the survey: 32% of respondents said that they used antibiotics to kill microbes (n = 111), 20% to treat pain, 15% to treat a sore throat, and 14% to treat respiratory tract infections (e.g., flu, pneumonia). There was a modicum of responses to options such as diarrhoea (5%), after delivery (4%) and malaria (2%).

> "It [Ampicillin] is good medicine such as you have femininity issue [infertility], body pain and throat pain it cures them I have good memories of that medicine." (Female patient, 45 years)

Patients reported using antibiotics after deliveries and during menstruation, since during these times a woman's body is considered to be susceptible to danger and infections. Losing blood was thought to weaken the body, and antibiotics were used to make the woman "feel good".

> "Those [Ampicillin] were for bleeding after delivery, that my parts of the body were open at that time, for this reason they [doctor and family] gave me medicine. [. . .] It [the antibiotic]

is good for our throat pain and period [menstruation] of my daughters, I have two young daughters at home, one of them is engaged, when we take this [Ampicillin], we feel ok." (Female patient, 45)

There were a few instances of patients using antibiotics after delivery (n = 4, 1%) or for bleeding issues (n = 8, 2%) in the 'other' option for the question "What do you use antibiotics for?". Interviews with health promoters confirmed this practice of using antibiotics after delivery:

"People take antibiotics for their wound infections and the women who delivered they take antibiotics because they have wounds and sores, so they take antibiotics." (Male HP, 40 years)

**Antibiotic prescription adherence and leftovers.**   Eighty-seven percent of survey participants indicated that doctors/dispensers explained that the drugs needed to be finished (n = 305), and 94% indicated that the instructions given on how to take the drugs were always clear (n = 331). That said, patients and caretakers stated in the interviews that they would stop taking the medication if they felt cured, discontinuing their prescribed pills even though the package or prescribed amount was unfinished. This was due to the belief that there was no need to take medication when feeling healthy–some even feared that it could harm them.

". . . medicine can be in situations that we need to take it; sometimes medicines can be harmful for you, for example when you are sick it may cure you but when you are not sick it can damage your body." (Male caretaker, 40 years)

Patients would use leftovers mainly for the recurrence of the same ailment, with the dosage reduced for children. Patients who could not afford antibiotics did not purchase the full course of treatment or stopped as soon as they felt cured in order to be able to store the leftovers for the next course of illness.

Twenty-three percent of survey participants responded that they sometimes ceased taking antibiotics early (n = 82). Of those who reported not finishing the full course of antibiotics, the main reason given was that the respondent reported feeling better/cured (85%, n = 70). In terms of the medication remaining, they either kept the antibiotics for future use (45%, n = 37), or threw them away (56%, n = 46).

Thirty-five percent of participants reported having antibiotics at home (n = 122). Of those participants, the most common reason they had antibiotics at home was because they had reserved them from an uncompleted former treatment (89%, n = 108). From the 122 participants that had antibiotics at home, the majority of participants responded they would use those antibiotics for the condition for which they were prescribed (85%, n = 104).

## Discussion

This mixed-methods study from Afghanistan explores the knowledge, perceptions and use of antibiotics among patients, medical prescribers and pharmacists. A limited understanding of antibiotics by patients and caretakers was observed in both the interviews and survey results, along with inappropriate use of antibiotics and pressure from patients demanding antibiotics. While various studies from both high and low- to middle-income countries report poor knowledge and inappropriate use of antibiotics among patients, the general population and prescribers [5–8, 37–40], this study reports findings from an underreported context: a hospital setting in a peri-urban environment in Kabul, Afghanistan. Findings highlight important

implications for public health policy and medicine supply regulations, patient and provider education, and clinical practice.

In general, antibiotics were commonly recognized by both healthcare workers and patient/caretaker participants, often with colloquial names relating to their perceived use in removing dirt and microbes quickly from the body, in response to local beliefs around living in a dirty, polluted environment. A similar theme of health associated with cleanliness in Afghanistan was found by Monaghan [41], with microbes coming from "dirty places" [p67]. A study in West Bengal, India found that patients' demanded antibiotics due to unhygienic environment and living conditions [10]. Our findings indicate that antibiotic use was characterized by people's perception of needing strong medication to counter physical and emotional distress in a dirty and polluted environment. Antibiotics appeared to have power even before they were used. Even the idea of getting an antibiotic, with its "innate" healing power, made it a highly desirable medication. "Only when it is consumed does the substance become a medicine. 'Wrong use' may render the best medicine useless or dangerous" [42]. In this study the symbolic value placed on antibiotics lies in this presumptive association with a powerful ability to not only cure but even *prevent* disease.

The same study from West Bengal also found patients demanded a quick cure [10]. In a conflict setting such as Afghanistan where people face social disadvantage and insecurity, a 'quick' medication is considered an important asset. The belief that antibiotics cure illnesses 'quickly' was observed in both the qualitative and quantitative findings of this study. Moreover, patients reported that they would switch healthcare providers when the treatment was perceived as being too slow.

In the search for 'quick' relief, patients and caretakers engaged in self-medication, often purchasing medications from private pharmacies. Private drug stores and drug vendors working in pharmacies are a major driver of antibiotic overuse in low-income countries [24]. Many studies have found that people are more likely to go straight to the pharmacy instead of consulting a doctor for medication when it is more accessible and when medical consultations are not easily available or affordable, including in Afghanistan [43–48]. Both our quantitative and qualitative findings confirm that patients obtained medications, including antibiotics, from elsewhere if the doctor did not prescribe them during their clinic visit. In private drug stores, where medicines supply are largely unregulated, untrained drug vendors lack the knowledge, training, and ethics around appropriate antibiotic use. Therefore, drug dispensers yield considerable influence on community drug use [42, 49–51].

Central to the patient's rationale is to get a prescription that contains many different drugs, with the more the better. Patients prioritize both quality and quantity, and expect a variety of 'many drugs' to be prescribed to have them ready when needed [24]. The act of prescribing multiple pills confirms for the patient that the doctor understands the extent of their suffering [40]. Antibiotics were not only used to treat illnesses but also to protect, purify and strengthen the body. It has been well documented in various contexts and countries that antibiotics are viewed as having the ability to prevent diseases [24]. Receiving drugs marks an important step in the consultation process–its end. Not receiving any drugs would mean for the patient that "it can't be all that serious if the doctor didn't prescribe antibiotics" [46]. However, both qualitative and quantitative findings suggest that patients acknowledged that they should abide by a doctor's prescription.

While doctors highlighted the expectations of patients to receive antibiotics in the qualitative interviews, the quantitative survey found that only a minority of participants expected to receive an antibiotic on the day they completed the survey at the hospital. However, participants generally expected to receive some sort of medication (if not antibiotics specifically), similar to a study on the misuse of antibiotics in nine different countries [23]. Patient

expectations of receiving drugs put pressure on doctors, who appeared to have conflicting obligations both to abide by MSF antibiotic prescription rules in the hospital and to prescribe generously in their private clinics to meet patient expectations. Various studies confirm that when patients expect to be prescribed an antibiotic, they receive it and when the doctor thinks the patient expects an antibiotic he or she is more likely to prescribe one [52–54]. Thus doctors' decisions on prescribing antibiotics are influenced not only by biomedical knowledge, training, and prescribing guidelines, but also by social, economic, and cultural factors [42]. Doctors struggle on a daily basis: on the one hand there is external pressure to appropriately prescribe fewer antibiotics (a long-term goal) but on the other hand there are daily encounters with patients and caretakers expecting to receive antibiotics. In treating patients "holistically," practitioners may prescribe antibiotics for sore throats [55].

Reduced antibiotic prescribing along MSF guidelines resulted in patient accusations that antibiotics were being reserved for others. However, patients still reported purchasing medications from drug stores, thus bypassing antibiotic prescribing guidelines followed by doctors at ASB hospital. Similar results were found in a study in northern Tanzania where the most prevalent form of irrational antibiotic use was non-prescription usage [39]. In Kabul, the practice of self-medication with antibiotics is common. Roien et al. surveyed visitors to private pharmacies in Kabul in 2017 and found 73% of participants had self-medicated with antibiotics in the past 12 months [56]. Negarandeh et al. surveyed visitors to the Central Polyclinic of Kabul in 2019 and found 35% of their participants self-medicated with antibiotics in the past 12 months [47]. In both studies the findings indicate that the understanding of appropriateness and lack of access to healthcare are similar issues influencing antibiotic use as found in our study.

In both the qualitative and quantitative findings, patients reported not completing courses of antibiotics, usually because they felt better, while keeping spares for themselves for later illnesses or for relatives. Failure of antibiotics to cure an illness was not attributed to antibiotic resistance but either to side effects or the need for additional stronger medication, as reported elsewhere [23, 39].

This study has several strengths and limitations. The mixed-methods design of this study allowed for a rich and in-depth exploration of key aspects of antibiotic use among patients, medical prescribers and pharmacists. Although the sample recruited was representative of a hospital OPD population in relation to demographic characteristics (i.e., age groups, ethnic backgrounds, gender), it may not be broadly representative of the general population, given the study sample was more likely to comprise people able to access and engage in healthcare. The field study team (anthropologist, translators and fieldworkers) were identified as MSF staff so they may have been perceived by participants as expecting responses that aligned with MSF prescribing policies, hence potentially introducing social desirability bias. To address such biases, the role of the anthropologist and fieldworkers, their neutrality, and the strict assurance of anonymity and confidentiality were carefully explained to all participants. As the study team was restricted to the ASB Hospital premises, interviews could not be conducted in the community. Community data collection could have enabled a better understanding of living conditions of patients and caretakers and been a more neutral territory to further reduce reporting biases. The views and knowledge of both staff and patients/caretakers were likely to differ from those in public health facilities because of the requirements of staff to follow MSF antibiotic prescribing guidelines, thus the findings may have limited generalizability.

The nature of the data sources has several limitations. Qualitative data are only a reflection of reality, as they are influenced by the presence of the researcher. In addition, the background of the researcher (gender, age, social status, origin etc.) shapes the research process. The researcher was aware of this effect and took a critical stance towards her position in the data gathering process and while analysing the findings. Both qualitative and quantitative data were

self-reported and were likely to have been affected by social desirability bias, so some antibiotic views or uses may have been inaccurately reported.

In 2015, the World Health Assembly endorsed the WHO Global Action Plan on Antimicrobial Resistance [57], and the Afghanistan National Action Plan on Antimicrobial Resistance 2017–2021 [58] has been developed. These action plans outline key objectives, strategic priorities and key actions to address antimicrobial resistance, including research and surveillance of antimicrobial resistance, improving awareness and understanding of antimicrobial resistance through effective communication, education and training, and optimization of antimicrobial use in human and animal health. Indeed, the diverse, powerful socio-cultural factors highlighted in our study influencing antibiotic views need to be considered when formulating public health strategies to shift attitudes and behaviours towards antibiotic use and tackle antibiotic resistance, including public health promotion and awareness raising, enhanced training and education of pharmacists, doctors, and drug vendors, advocacy for policy and medicine supply regulation, and antibiotic stewardship programs in clinical settings. Our findings add to the evidence-base for understanding of access to and public perception of antibiotics.

## Conclusions

This study showed that, among outpatients attending ASB Hospital in Kabul, the desire to receive and use antibiotics had an important social and symbolic meaning. Public perceptions of antibiotics as powerful, fast-acting, and a way to improve many conditions for which they were not indicated led to a high demand to receive them. If antibiotics were not obtained from the doctor on prescription, they would be purchased on the open market. The frequency with which antibiotic treatment courses were not completed, with spares kept for future use, adds to the issues around inappropriate antibiotic use. These findings highlight the urgent need for awareness raising, education and training of the general public, doctors and pharmacists, regulation of antibiotic dispensing in private pharmacies, and antibiotic stewardship programs in hospitals.

## Supporting information

**S1 Appendix. Interview question guide.**
(PDF)

**S2 Appendix. Questionnaire in English and Pashto.**
(PDF)

**S1 Table. Responses to survey questions by patients and caretakers at ASB District Hospital, Kabul, Afghanistan, between October 10 and November 11, 2014.**
(DOCX)

## Acknowledgments

The Authors would like to thank the ASB staff and patients who have generously participated in the study, and Dr Anthony Reid and Ms Caroline Walker for their editorial advice.

## Author Contributions

**Conceptualization:** Doris Burtscher, Rafael Van den Bergh, Masood Nasim, Gbane Mahama, Sokhieng Au, Catherine Van Overloop, Sahar Bajis.

**Data curation:** Doris Burtscher, Masood Nasim, Gbane Mahama.

**Formal analysis:** Doris Burtscher, Rafael Van den Bergh, Anita Williams, Abdul Sattar, Suzanne Penfold, Sahar Bajis.

**Investigation:** Doris Burtscher.

**Methodology:** Rafael Van den Bergh, Masood Nasim, Gbane Mahama, Sokhieng Au, Sahar Bajis.

**Supervision:** Doris Burtscher.

**Validation:** Doris Burtscher, Rafael Van den Bergh, Anita Williams, Catherine Van Overloop.

**Writing – original draft:** Doris Burtscher, Sahar Bajis.

**Writing – review & editing:** Doris Burtscher, Rafael Van den Bergh, Masood Nasim, Gbane Mahama, Sokhieng Au, Anita Williams, Abdul Sattar, Suzanne Penfold, Catherine Van Overloop, Sahar Bajis.

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
