## [Decision Letter · Decision Letter 0]

15 Jun 2021

PONE-D-21-14869

‘They eat it like sweets’: A mixed methods study of antibiotic perceptions and their use among patients, prescribers and pharmacists in a district hospital in Kabul, Afghanistan

PLOS ONE

Dear Dr. Burtscher,

Thank you for submitting your manuscript to PLOS ONE. After careful consideration, we feel that it has merit but does not fully meet PLOS ONE’s publication criteria as it currently stands. Therefore, we invite you to submit a revised version of the manuscript that addresses the points raised during the review process.

The manuscript is well-written and covers an important topic from a region of the world which is not often studied. Kindly consider the suggestions to further strengthen the manuscript and submit a  revised version. 

We look forward to receiving your revised manuscript.

Kind regards,

Pathiyil Ravi Shankar

Academic Editor

PLOS ONE

Journal Requirements:

Additional Editor Comments (if provided):

This is an important and interesting study from a region of the world which is not often studied. I enjoyed reading the manuscript. My suggestions for further improvement are:

I am not aware of the situation on the ground in Afghanistan. A factor of concern to me is the fact that there may be only two Afghans as co-authors in this manuscript. Is there any reason for this lesser number? I am not aware of the involvement of Afghans in their countries healthcare system and I also know that MSF may depend more on volunteer doctors.

The study was conducted in 2014 and there is a long gap between the conduct of the study and the submission of the manuscript in 2021. The difficult situation in Afghanistan may have contributed though I am not sure.

There is a lack of clear description of how the findings from the qualitative study were used to develop the questionnaire. There is no clear demarcation between the qualitative and quantitative phase of the study.

The authors have not mentioned how the data from the questionnaire was analysed.

What exactly is meant by a health promoter?

Are the medical staff Afghans or expatriates? My understanding on reading the manuscript is that the doctors working in the hospital also have their private practice. I am not sure if this is correct.

The findings of the study point to a disturbing situation which is like that seen in other deprived areas. The authors have described the strengths and limitations of the study clearly. Considering the limitations of the current study do the authors recommend further studies in other regions of the country, though, this may be difficult.

In the supplementary file question 4, what is meant by serum?

This is an important study and I assume the situation may not have changed substantially in the seven years since the study was conducted.

Reviewers' comments:

Reviewer's Responses to Questions

**Comments to the Author**

1. Is the manuscript technically sound, and do the data support the conclusions?

Reviewer #1: Yes

2. Has the statistical analysis been performed appropriately and rigorously? 

Reviewer #1: I Don't Know

3. Have the authors made all data underlying the findings in their manuscript fully available?

Reviewer #1: No

4. Is the manuscript presented in an intelligible fashion and written in standard English?

Reviewer #1: Yes

5. Review Comments to the Author

Reviewer #1: Authors have to revise the manuscript as per the comments and suggestions given. The data is a bit old and is of 2014. How was the questionnaire for the quantitative research structured and validated? Authors can also compare their results with the current scenario making some recommendations.

6. PLOS authors have the option to publish the peer review history of their article (what does this mean?). If published, this will include your full peer review and any attached files.

Reviewer #1: No

---

## [Author Response · Author response to Decision Letter 0]

17 Aug 2021

Response to academic editor and reviewer(s)

Thank you very much for considering a revised manuscript. We have addressed the reviewer comments and made revisions to the manuscript. Please see below our response to the reviewer comments in italics (with page numbers pertaining to tracked-manuscript).

Journal Requirements:

Thank you for this recommendation, we have reconsulted the journal’s formatting guidelines and have changed the manuscript accordingly. 

We will submit the questionnaire used in the quantitative part of the study as additional information. The questionnaire is one document in Pashto and English language. Additionally we will submit the qualitative question guide. We will also upload the survey data as a supplementary file. 

Important: If there are ethical or legal restrictions to sharing your data publicly, please explain these restrictions in detail. Please see our guidelines for more information on what we consider unacceptable restrictions to publicly sharing data: 

http://journals.plos.org/plosone/s/data-availability#loc-unacceptable-data-access-restrictions. Note that it is not acceptable for the authors to be the sole named individuals responsible for ensuring data access.

We have provided the quantitative data as a supplementary file. The qualitative data cannot be shared due to the following ethical reasons: 

- Even though de-identified, transcripts came from a vulnerable and restricted group of respondents and the narratives there contained might disclose personal aspects of their life. In this we follow our ERB protocol for protecting participants

- In the Informed consent form we did not indicate the possibility for public access to their narratives; it was only stated that these will be used for analysis only by the research group – again following ERB protocol and that quotes will be used anonymized in a publication. 

4. A factor of concern to me is the fact that there may be only two Afghans as co-authors in this manuscript. Is there any reason for this lesser number? I am not aware of the involvement of Afghans in their countries healthcare system and I also know that MSF may depend more on volunteer doctors.

Thanks for this important comment. We agree with the reviewer that there is an underrepresentation of Afghan co-investigators in this study, however; our two Afghan colleagues and co-authors on this manuscript have made substantial contributions to the design, implementation of the study and interpretations of findings. In addition, an Afghan pharmacy assistant was trained to conduct data entry for the study as part of research skills building in the hospital. MSF continues to take measures to support and provide training to local staff to lead research projects. It should be noted that MSF employs local doctors in accordance with local requirements – they are not volunteers. 

5. The study was conducted in 2014 and there is a long gap between the conduct of the study and the submission of the manuscript in 2021. The difficult situation in Afghanistan may have contributed though I am not sure.

While there has been an extended period of time between the conduct of this study and submission of manuscript, the nature of our findings remains representative of the community, relevant and applicable to the current context and issue of antimicrobial use and resistance and has important policy and practice implications.

6. There is a lack of clear description of how the findings from the qualitative study were used to develop the questionnaire. There is no clear demarcation between the qualitative and quantitative phase of the study.

We have added text in the “Quantitative component” of the methods section outlining how the findings from the qualitative phase was used to develop the survey. We have also added time periods for when the two phases were conducted in order to delineate between the two phases of the study. Please see page 4 and page 7. 

7. The authors have not mentioned how the data from the questionnaire was analysed.

We have provided further detail in the methods section to clarify how this analysis was conducted. Please refer to page 8 in the manuscript. 

8. What exactly is meant by a health promoter?

The role of health promoters is to facilitate and deliver health messages and education to patients/communities around disease prevention and control. They play an important role in ensuring health services are accessible and relevant to communities. In the Ahmed Shah Baba hospital in Kabul health promotion activities included providing information to caretakers in the male and female waiting area on antibiotic characteristics, and antibiotic resistance, and for which ill-health conditions antibiotics should be used. Additionally, people were sensitised about mental health services provided by MSF. 

9. Are the medical staff Afghans or expatriates? My understanding on reading the manuscript is that the doctors working in the hospital also have their private practice. I am not sure if this is correct.

At the time of the study, there were three international medical staff and 215 national medical staff employed by MSF at the ASB hospital. Additional to this there were 43 medical Ministry of Public Health (MoPH) staff supported by MSF working at the hospital. In accordance with local human resources regulations, national staff were entitled to practice in private clinics outside of their employment at the hospital. 

10. The findings of the study point to a disturbing situation which is like that seen in other deprived areas. The authors have described the strengths and limitations of the study clearly. Considering the limitations of the current study do the authors recommend further studies in other regions of the country, though, this may be difficult.

Ideally, we would repeat the study in our other projects, and we are in discussions with the team in Afghanistan regarding the possibility of doing a similar study to update and extend our knowledge on this topic. That said, considering the current security context this idea may be delayed. 

11. In the supplementary file question 4, what is meant by serum?

Thank you for this clarifying question. In the local context, serum means medication received through an intravenous line. 

12. This is an important study and I assume the situation may not have changed substantially in the seven years since the study was conducted.

We agree with the reviewer that despite the length of time since the conduct of study, the findings are still relevant and important in addressing knowledge gaps and policy implications in Afghanistan and the regions more broadly.

The following responses are from comments made in the pdf of our article

1. Are expired medicines also circulated?

We are not aware of anything documented but this is possible; it would be outside of the scope of this paper to discuss this subject. 

2. This can be rewritten in a simple language.

We have made changes to that sentence to clarify the language. Please see on page 3 in the manuscript. 

3. What was the basis for 30% prevalence?

The 30% prevalence assumption was based upon findings from similar studies. 

4. How was the questionnaire validated?

The data was double-entered into EpiData and cross-checked between the two entries. This is what was meant by validated. The manuscript has been changed to say “cross-checked” to clarify this. Additionally, text has been added in line 194 that the questionnaire was pre-piloted in the population. Please refer to page 8.

---

## [Decision Letter · Decision Letter 1]

13 Sep 2021

PONE-D-21-14869R1‘They eat it like sweets’: A mixed methods study of antibiotic perceptions and their use among patients, prescribers and pharmacists in a district hospital in Kabul, AfghanistanPLOS ONE

Dear Dr. Burtscher,

Thank you for submitting your manuscript to PLOS ONE. After careful consideration, we feel that it has merit but does not fully meet PLOS ONE’s publication criteria as it currently stands. Therefore, we invite you to submit a revised version of the manuscript that addresses the points raised during the review process.

Kindly revise the manuscript as recommended by the reviewer for further consideration by the journal. 

We look forward to receiving your revised manuscript.

Kind regards,

Pathiyil Ravi Shankar

Academic Editor

PLOS ONE

Journal Requirements:

Additional Editor Comments (if provided):

Reviewers' comments:

Reviewer's Responses to Questions

**Comments to the Author**

1. If the authors have adequately addressed your comments raised in a previous round of review and you feel that this manuscript is now acceptable for publication, you may indicate that here to bypass the “Comments to the Author” section, enter your conflict of interest statement in the “Confidential to Editor” section, and submit your "Accept" recommendation.

Reviewer #1: All comments have been addressed

Reviewer #2: (No Response)

2. Is the manuscript technically sound, and do the data support the conclusions?

Reviewer #1: Yes

Reviewer #2: Partly

3. Has the statistical analysis been performed appropriately and rigorously? 

Reviewer #1: I Don't Know

Reviewer #2: (No Response)

4. Have the authors made all data underlying the findings in their manuscript fully available?

Reviewer #1: Yes

Reviewer #2: (No Response)

5. Is the manuscript presented in an intelligible fashion and written in standard English?

Reviewer #1: Yes

Reviewer #2: Yes

6. Review Comments to the Author

Reviewer #1: The authors have answered to the queries made by the reviewers. The manuscript now reads well and is more robust than before.

Reviewer #2: The authors deserve appreciation for having studied an important topic. The following queries need to be clarified:

1. (a) Did any of the provider-respondents receive (uniform) training in rational use of antibiotics? (b) If yes or no, please mention that explicitly in the methods section. (c) If yes, how many, and whether data for those trained and not trained were analyzed? How was this point captured in the questionnaire?

2. Was participation in training a criterion for inclusion / exclusion of provider-respondents in the study?

3. Were user-respondents given health education on the use of antibiotics?

4. (a) What was the rationale behind choosing this hospital for data collection? (b) Why was it restricted to just one?

5. In the tables, indicate as to how many respondents were there below age 20 years.

6. (a) Table 2: age not recorded is 41! Not even rounded off values available? (b) How was age determined?

Answer to most of these questions should appear in the methods section.

7. PLOS authors have the option to publish the peer review history of their article (what does this mean?). If published, this will include your full peer review and any attached files.

Reviewer #1: No

Reviewer #2: No

---

## [Author Response · Author response to Decision Letter 1]

29 Oct 2021

Response to journal editor and reviewer comments

Thank you very much for considering the revised manuscript. We have addressed the reviewer comments and made revisions to the manuscript. Please see below our response to Reviewer #2 comments in italics (with line numbers pertaining to tracked-manuscript).

1. (a) Did any of the provider-respondents receive (uniform) training in rational use of antibiotics? (b) If yes or no, please mention that explicitly in the methods section. (c) If yes, how many, and whether data for those trained and not trained were analyzed? How was this point captured in the questionnaire?

Thank you for this question. In general, medical staff receive education and training on MSF clinical guidelines and good prescribing practices for most common clinical conditions, including rational use of antibiotics. However, a uniform training or intervention focused on rational use of antibiotics was not implemented during the study period. In addition, we did not have information on which medical doctors received any training and an evaluation as such is outside the scope of this study. Note, the questionnaire was only administered to patients and caretakers. We have revised the manuscript to clarify this point (lines 103-7). 

2. Was participation in training a criterion for inclusion / exclusion of provider-respondents in the study?

As stated above, a uniform training was not implemented during the study period and such was not applicable in terms of eligibility criteria.

3. Were user-respondents given health education on the use of antibiotics?

As part of standard health promotion activities, education on antibiotic use was provided to patients and caretakers in the male and female waiting areas of the hospital. This activity has been described in the “Settings” paragraph (lines 109-11). No additional education sessions or interventions related to this were provided to participants. 

4. (a) What was the rationale behind choosing this hospital for data collection? (b) Why was it restricted to just one?

As described in the manuscript, the study was conducted on the basis of anecdotal evidence of inappropriate antibiotic use in the ASB OPD, and the gaps in knowledge regarding antibiotic use in Kabul and Afghanistan at large. Findings from this study have implications for intervention, education and training of medical staff and the community. ASB hospital was also chosen as it was the only MSF project (at the time) with busy outpatient and inpatient departments with a variety of specialties including paediatrics, sexual reproductive health, and surgery. 

5. In the tables, indicate as to how many respondents were there below age 20 years.

In both Tables 1 and 2 the number of participants aged under 20 have been listed; there were six (6) participants aged under 20 in the interviews, and 53 participants surveyed. 

6. (a) Table 2: age not recorded is 41! Not even rounded off values available? (b) How was age determined?

We are unsure of this comment from Reviewer #2 regarding the values of the age not recorded. There were four (4) individuals that did not provide age, which made up 1%. The participants may not have provided their age because they did not want to, did not know their age, or may be an error of data encoding. We feel that this very small number of missing data does not impact the quality of analysis or findings.

---

## [Editor Report · Decision Letter 2]

3 Nov 2021

‘They eat it like sweets’: A mixed methods study of antibiotic perceptions and their use among patients, prescribers and pharmacists in a district hospital in Kabul, Afghanistan

PONE-D-21-14869R2

Dear Dr. Burtscher,

We’re pleased to inform you that your manuscript has been judged scientifically suitable for publication and will be formally accepted for publication once it meets all outstanding technical requirements.

Kind regards,

Pathiyil Ravi Shankar

Academic Editor

PLOS ONE
---

## [Editor Report · Acceptance letter]

10 Nov 2021

PONE-D-21-14869R2 

‘They eat it like sweets’: A mixed methods study of antibiotic perceptions and their use among patients, prescribers and pharmacists in a district hospital in Kabul, Afghanistan 

Dear Dr. Burtscher:

I'm pleased to inform you that your manuscript has been deemed suitable for publication in PLOS ONE. Congratulations! Your manuscript is now with our production department. 

Kind regards, 

on behalf of

Dr. Pathiyil Ravi Shankar 

Academic Editor

PLOS ONE